# OpenReview forum: "LLM Agents can Autonomously Hack Websites"
_TMLR — Rejected by TMLR_

### Review · Reviewer_hxGJ · 2024-05-01

**Summary Of Contributions:**

This papers proposes a methodology of leveraging LLMs to hack websites.

**Audience:**

No

**Claims And Evidence:**

Yes

**Requested Changes:**

Need to clarify the research contributions of the work.

**Strengths And Weaknesses:**

It is great to see that authors is able to engineer a method to hack website including interfacing the LLMs with websites, giving them access to documents and planning (which I did not find the details how the planning works). However, I did not see any novelty of the work from the scientific stand point. The authors did not use LLMs to discover any new TYPE of vulnerability of website. Given the current version of the manuscript, I am not even sure the issues discovered can not be discovered by existing vulnerability detection techniques (e.g. program verification, penetration testing).

Overall, it might be useful engineering work but i am not sure this is worthy a publication at a research journal.

---

> ### Author Response · Authors · 2024-05-08
>
> Dear reviewer,
>
> Thank you for your comments on our experiments. We would like to clarify several points around our scientific contributions.
>
> 1. As the other reviewers have pointed out, we have conducted experiments to determine _which_ models can perform exploits, showing a potential “emergent capability.” We believe that this is valuable from a scientific point of view.
> 2. We compare GPT-4 to an open-source vulnerability scanner, ZAP (Section 5.3). ZAP can detect 8 out of the 15 vulnerabilities, compared to 11 out of the 15 by GPT-4. Thus, GPT-4 is already more capable than ZAP on our benchmark. Furthermore, we are unaware of _autonomous_ penetration testers.
> 3. Many (if not the majority) of real-world vulnerabilities are “known” vulnerabilities, such as XSS attacks or SQLi attacks. For example, one of the [largest cybersecurity attacks of all time](https://www.upguard.com/blog/cve-2021-26855) exploited an SSRF attack, a known vulnerability. Thus, we believe that finding known vulnerabilities is incredibly important.
> 4. Beyond the existing models, we believe that our work sheds light on the potential of future trends. GPT-4 is already more capable than GPT-3.5 and all of the open source models we tested. We anticipate that future models will be even more capable, which our work provides evidence for.
>
> Thank you for your consideration,
>
> Paper 2537 authors

---

### Review · Reviewer_d4ZH · 2024-05-03

**Summary Of Contributions:**

Contributions:
1. The authors found GPT4 can autonomously hack websites without needing to know the specific vulnerability beforehand.
2. Open LLMs tested in the paper cannot hack websites at all (without specifying the specific vulnerability). Therefore, such autonomous hacking capability is "emergent" only in frontier LLMs.

**Audience:**

Yes

**Broader Impact Concerns:**

No concerns. No specific prompts or codes are released in the paper to instruct how to perform autonomous website hacking.

**Claims And Evidence:**

No

**Requested Changes:**

1. (required) Please run open LLMs for many more trials and report the success rate. If the authors believe this is not necessary, please provide justifications (ideally in the updated paper).
2. (required) Please highlight the setup difference in Table 2 and Table 4, so that the numbers are not confusing (contradicting).
3. (required) Please tone down the claim that open LLMs cannot do autonomous hacking at all (c.f. the weakness I mentioned earlier)

**Strengths And Weaknesses:**

Strength:
* To my best knowledge, this is the first paper that systematically study LLM agents' capability in cybersecurity attacks.

Weakness:
* Writing can be improved, e.g. for better clarity. For example, Table 2 shows that all open LLMs have 0% success rate on autonomous hacking. Table 4 show the success rate for each vulnerability for GPT-4 and OpenChat3.5, where OpenChat3.5 has positive success rate on several tasks. This is a bit confusing on the first read. My understanding is, table 4's setup requires specifying the specific vulnerability.
* Potential flawed experimental methodology:
   * If I understand table 4 correctly, OpenChat3.5 does have capability for certain types of hacks (if the vulnerability is specified). If so, can we simply use the agent and enumerate all 15 vulnerabilities for trial-and-error? I find the claim that (tested) open LLMs cannot autonomously hack website at all is over-claimed.
   * Each agent is given a budge of 5 trials, with each trial under 10 minutes of execution. I understand the budget constraint for using OpenAI's API. It seems important to try many more trials for open LLM models to see if it can "get lucky". As the authors mentioned, for cybersecurity, only one successful hack is enough. While this is not an apple-to-apple comparison, it serves as a "stress test". Without enough trials, the claim on these open LLMs lack autonomous hacking capability is lacking support.

---

> ### Author Response · Authors · 2024-05-08
>
> Dear reviewer,
>
> Thank you for your thoughtful review. We respond to your comments below and will update our manuscript once our experiments are done.
>
> ## Improving the description of Table 2 vs Table 4
>
> We will change the terminology to be “exploit” and “detection.” A successful _detection_ is when the agent determines the correct type of exploit to try (e.g., XSS vs SQLi). We note that a successful detection may be a hallucination, and even after a successful detection, the model may not be able to exploit the vulnerability. A successful _exploit_ is when the agent correctly carries out the exploit (e.g., exfiltrated the data via an SQLi).
>
> Table 2 shows the exploit rate, which is the more difficult task. Table 4 shows the detection rate. We will update our manuscript to clarify this point.
>
>
> ## Understanding OpenChat’s performance
>
> OpenChat cannot exploit the websites even when the vulnerabilities are specified. It can determine the correct type of vulnerability to try in some cases. However, these successful detections can be hallucinations, and even after a successful detection, the agent may not be able to perform the exploit.
>
> In the given ten runs that I ran it did say successful exploitation but that was hallucination as I didn’t really see or get an output. It just said that it successfully exploited the site. This could also be due to a potential update in the model
>
>
> ## More trials on open source models
>
> Many of the models produce incoherent output (e.g., repeated strings) or incorrect tool use. Furthermore, we manually evaluated each of the runs to determine whether or not an exploit was successful. As such, we will run more trials of OpenChat and report results once the experiments conclude.
>
>
> Thank you for your consideration,
>
> Paper 2537 authors

---

> > ### Comment · Reviewer_d4ZH · 2024-05-08
> > **followup**
> >
> > Thank you for your responses, especially the exploit vs detection clarification.
> >
> > A few questions:
> > 1. How can you tell if a successful detection is a hallucination or not? Is this defined by being able to exploit the detected vulnerability or something else? Could you please provide/describe some examples?
> > 2. You mentioned "Many of the models produce incoherent output (e.g., repeated strings)". It might be useful to include some examples in the paper as additional evidence and insights. I wonder how much of this is due to agent programming 'bugs' (e.g. misuse of the LLM's API) or inherent issues with the LLMs.

---

> > > ### Author Response · Authors · 2024-05-15
> > >
> > > Thank you for your questions and reading our manuscript and responses carefully.
> > >
> > > 1. Hallucinations in this setting can be difficult to disentangle. However, open-source models: 1) claim a vulnerability exists when it does not (e.g., claiming an SQLi exists when an XSS does not), and 2) cannot carry out the exploit even when it claims the correct vulnerability exists. Thus, we strongly believe that the open-source models are hallucinating.
> > >
> > > 2. We believe it is highly likely that open-source models will improve (with strong evidence already from Llama-3), including in its tool use. We hope that our work inspires future work to investigate improved models for cybersecurity applications, but view this as outside the scope of the present work.
> > >
> > > Best,
> > >
> > > Paper 2537 authors

---

### Review · Reviewer_hQZd · 2024-05-03

**Summary Of Contributions:**

The paper demonstrates how large language models (LLMs) agents can hack websites when provided access to general documentation, tools for function calling and the feedback/history of the agent's interaction with tools and websites. The paper explores a list of vulnerabilities with varying difficulty levels and showcases that GPT-4 the most capable agent is able to autonomously hack websites followed by GPT-3.5. On the contrary, open source models are incapable of using tools correctly, fail to plan appropriately thus failing to hack websites. Overall the paper highlights the risk of using LLMs with tools and function calling and how cost effective it is to use LLM APIs to perform attacks.

**Audience:**

Yes

**Broader Impact Concerns:**

The paper has a Broader Impact Statement section about there work.

**Claims And Evidence:**

No

**Requested Changes:**

Mentioned in Weaknesses above.

**Strengths And Weaknesses:**

**Strengths**

**Important and Novel Problem**

1. The paper tackles an important and novel problem concerning the potential cybersecurity threat posed by LLM agents when paired with tools and function calling mechanisms.
3. The paper studies a variety of vulnerabilities with varying difficulty levels.
4. The results in the paper highlights the gap between open and closed source models in skills like function calling, planning, tool use, etc.

**Good Experiments**

1. The paper has important results demonstrating that GPT-4 can successfully hack websites that require upto 44 function calls and is able to use the tool feedback to backtrack and plan accordingly.
2. It has ablations showing the importance of providing the documentation and appropriate prompt for attack to succeed.
3. It has a comparison with an open-source security tool to show that GPT-4 is more capable then openly available security software.
4. The authors do a cost analysis, highlighting how cheaper it is to use an LLM agent for hacking instead of using experts for exposing vulnerabilities.

**Weaknesses**

**No insights into why open source models are unable to hack websites**

1. There is no explanation and analysis of why the open source models are unable to do security hacks. The authors must have added some analysis showing where specifically open models fail, like how far they are able to go before failing or where they make mistakes or are they simply unable to use tools/function calling?
2. Another alternative to investigate the worse performance of open source models is to use a human in the loop setup that rectifies some generated outputs or explicitly instructs the model after certain steps to use a particular tool/function. Although this will not be fully autonomous, it will help in understanding the gap between GPT-4 and other open models in a better way.
3. Also the paper takes some popular open source models but those models might not be strong open models for this task. Why are there no open models used from recent research on tool use, function calling, LLM agents? As those models are trained for similar use cases, maybe they can understand the tools or functions better and hence can be better candidates for hacking?
4. Also how is the detection success rate differ from the attack success rate. The results in Table 4 needs more clarification.

**Details regarding the website setup and more evidence.**

1. A detailed description of the website setup in the sandbox environment would have been useful. What does the website contain, how was it set up, how does it have all the vulnerabilities listed in table 1.
2. There are also no examples in the paper to understand what GPT-4 is doing, what does the task look like, what is the input and output of the model at each step. What tools/functions call it does, etc. Due to this, the numbers in table 2-3 are hard to interpret. Although the authors cannot expose these details, a mock setup of some custom website and tools to just showcase some examples would help understand the GPT-4 behavior and interpret the results better. If a mock setup or any kind of example is not possible then a systematic human evaluation would help (described below).

**Human evaluation of the steps followed.**

1. The paper has results that GPT-4 uses upto 44 calls and can successfully hack websites and can even backtrack and plan accordingly. But there is no concrete evaluation of each step in the paper. There should be a human evaluation (by some expert) on each step followed by GPT-4 to see if it really follows the correct steps to arrive at the solution. Are the steps similar to what an expert hacker would do? Just observing if it is able to hack or not is not a good metric of success. In Math or reasoning tasks simply giving a correct answer is not enough, the steps followed to arrive at the answer should be correct as well. I think a similar evaluation setup could be very helpful here. This will highlight how deeply LLM agents understand vulnerabilities in cybersecurity.

---

> ### Author Response · Authors · 2024-05-08
>
> Dear reviewer,
>
> Thank you for highlighting the strengths of our work, particularly our existing experiments. We have responded to your weak points below and will update our manuscript once our experiments are done.
>
> ## Why are open source models are unable to hack websites?
>
> Before responding to the individual questions, we would like to clarify the purpose of our work.  In our work, we are focused on using widely available models to attempt website hacking. As such, we focused on models that are easily accessible via APIs. Thus, the open source models we evaluate on were highly ranked models on together.ai.
>
> 1. Many of these open source models produce nonsensical outputs (e.g., repeated strings) or are unable to effectively use tools (e.g., wrong tool usage, malformed outputs, etc.). For example, _every single run_ on the Llama-2 chat model has malformed outputs or incorrect tool usage. Furthermore, the majority of open source models are incapable of even determining the correct exploit to test. As such, we believe that these open source models are simply incapable of being used in complex agent environments.
> 2. Since the majority of the open source models fare poorly in the agent setting, we will compute statistics on malformed outputs and incorrect tool use.
> 3. Since we are focused on widely available models, we used models available from together.ai that were highly ranked in ChatBot arena.
> 4. We considered a model to have successful detection if it attempted the correct exploit (e.g., XSS vs SQLi) that the website was vulnerable to. However, a successful detection may be a hallucination and the open source models still cannot exploit the vulnerabilities after successful detections. We will make this point more clear in the paper.
> 5. The models also try to perform multiple attacks when there is clearly only one exploit to be done.
> 6. It also falsely detects successability when performing the exploits for exploits that are not even reasonable for the environment in place.
>
>
> ## Website setup and more evidence
>
> 1. We will update our paper with a more detailed description of each vulnerability.
> 2. The functions GPT-4 have access to are low-level functions to manipulate the Playwright browser testing framework. These functions include reading the code on the page (HTML and JavaScript) and clicking on elements. Thus, the number of actions is high because many functions must be called to simulate a single “human action.” We will showcase a full example in the paper.
>
>
> ## Human evaluation
>
> 1. To clarify our evaluation metric, we manually inspected every run to see if the model successfully hacked the website or not. Thus, all of our metrics are computed with human evaluations. GPT-4 does take actions similar to what an expert human would do, given the base functionality it has (e.g., reading the webpage, clicking on elements).
>
> Thank you for your consideration,
>
> Paper 2537 authors

---

### Decision · Action_Editor_47Yc · 2024-06-18

**Recommendation:** Reject

**Comment:**

The reviewers were not optimistic about the clarity of the contribution. One reviewer considered the results not surprising, but the issue of novelty is not crucial for the decision.
Most of the points brought up by the reviewers can be attributed to the lack of details or source code for this experimental paper.
If accepted, it'd be hard to compare with this submission.

Perhaps it would be possible to show the power of LLMs in a related task that is not about vulnerabilities.
Meanwhile, this AE doesn't see how a revised version of this paper could be resubmitted.

**Audience:**

Using LLMs for discovering website vulnerabilities is a topic of interest.

**Claims And Evidence:**

The paper studies whether using LLMs to detect security vulnerabilities in websites is possible, assuming humans can check the vulnerabilities afterwards. The paper considers a number of open LLMs and a closed LLM, GPT-4 by OpenAI. The paper describes the proposed method, saying that the LLM-based agents give the agents the ability to read documents, call functions to manipulate a web browser and retrieve results, and access context from previous actions.

Some reviewers had concerns about the details of the methods and the specific vulnerabilities. The paper was updated by adding the specific vulnerabilities. The paper argues that revealing additional details might pose a risk, so the paper does not includes enough details to reproduce the experiments. For instance, the reader is left without details about the "six documents that broadly cover a wide range of web attacks." The lack of details for reproducing the results of the paper makes it also hard to qualify the correctness of the contribution.

As the paper focuses on discovering availabilities, while the paper argues that additional details cannot be provided, we are left with claims that are hard to publish in an academic publication.